# Thymidine Kinase 1 Expression Correlates with Tumor Aggressiveness and Metastatic Potential in OSCC

**DOI:** 10.3390/diagnostics15121567

**Published:** 2025-06-19

**Authors:** Chia-Jung Lee, Pei-Wen Peng, Chia-Yu Wu, Tsung-Ming Chang, Ju-Fang Liu, Kuan-Chou Lin

**Affiliations:** 1Department of Otolaryngology Head and Neck Surgery, Shin-Kong Wu-Ho-Su Memorial Hospital, Taipei City 111045, Taiwan; m006578@ms.skh.org.tw; 2School of Medicine, Fu-Jen Catholic University, Taipei City 24205, Taiwan; 3School of Dental Technology, College of Oral Medicine, Taipei Medical University, Taipei City 111031, Taiwan; apon@tmu.edu.tw (P.-W.P.); m204097001@tmu.edu.tw (C.-Y.W.); a03441@tmu.edu.tw (T.-M.C.); 4Division of Oral and Maxillofacial Surgery, Department of Dentistry, Taipei Medical University Hospital, Taipei City 111031, Taiwan; 5Translational Medicine Center, Shin-Kong Wu Ho-Su Memorial Hospital, Taipei City 111045, Taiwan; 6School of Oral Hygiene, College of Oral Medicine, Taipei Medical University, Taipei City 111031, Taiwan; 7Department of Medical Research, China Medical University Hospital, China Medical University, Taichung City 404271, Taiwan; 8School of Dentistry, College of Oral Medicine, Taipei Medical University, Taipei City 111031, Taiwan; 9Department of Oral and Maxillofacial Surgery, Wan Fang Hospital, Taipei Medical University, Taipei City 111031, Taiwan

**Keywords:** oral squamous cell carcinoma (OSCC), thymidine kinase 1 (TK1), metastasis, biomarker, differentially expressed genes (DEGs)

## Abstract

**Background/Objectives:** Oral squamous cell carcinoma (OSCC) is the most prevalent malignancy of the oral cavity and is frequently diagnosed at an advanced stage, resulting in poor prognosis and limited treatment options. Identifying reliable biomarkers that can predict tumor progression and serve as therapeutic targets remains an urgent clinical need. **Methods:** To identify key molecular drivers in OSCC, we performed an integrative bioinformatics analysis of five OSCC-related microarray datasets from the Gene Expression Omnibus (GEO). Differentially expressed genes (DEGs) were identified and subjected to functional enrichment, protein–protein interaction (PPI) network construction, and hub gene ranking using Cytoscape. Candidate genes were further validated using TCGA, UALCAN, and the Human Protein Atlas. In vitro functional assays were performed to evaluate the effect of TK1 knockdown on cell migration. **Results:** A total of 138 common DEGs were identified across datasets. GO enrichment revealed that these genes were associated with cell proliferation, extracellular matrix organization, and metastasis-related processes. Thymidine kinase 1 (TK1) was identified as a key hub gene and found to be consistently overexpressed in OSCC tissues. Kaplan–Meier analysis showed that high TK1 expression correlated with poor overall survival in head and neck cancer. TK1 knockdown in OSCC cell lines significantly impaired cell migration and wound-healing ability. **Conclusions:** Our findings suggest that TK1 plays an active role in promoting OSCC progression and may serve as a prognostic biomarker and potential therapeutic target for metastatic OSCC.

## 1. Introduction

Oral squamous cell carcinoma (OSCC) accounts for over 90% of oral malignancies and is frequently diagnosed at advanced stages, resulting in a persistently poor 5-year survival rate of approximately 50%, despite advances in surgery, radiotherapy, and systemic therapy [1]. Among various prognostic factors, cervical lymph node metastasis is one of the most critical determinants of poor outcome in OSCC patients [2,3,4]. Therefore, identifying molecular markers that reliably predict metastatic potential is crucial for guiding treatment strategies and improving patient outcomes.

Thymidine kinase 1 (TK1) is a cytosolic enzyme involved in the salvage pathway of DNA synthesis and is tightly regulated by the cell cycle, peaking during the S phase. TK1 catalyzes the phosphorylation of thymidine into thymidine monophosphate, a key step in DNA replication. TK1 overexpression has been reported in a variety of malignancies, including colorectal, breast, lung, and esophageal cancers, where it is associated with advanced clinical stage, lymph node metastasis, and reduced overall survival [5,6,7,8,9]. In lung adenocarcinoma, TK1 has been shown to promote tumor progression through activation of Rho GTPase signaling and transcriptional regulation via the MAPK–MAZ axis [10]. Furthermore, elevated serum TK1 levels have been proposed as biomarkers for tumor burden, recurrence risk, and treatment resistance [5,7]. In addition, TK1 has been evaluated as a marker of tumor proliferation, primarily through immunohistochemistry and functional imaging using 18F-fluorothymidine (18F-FLT) positron emission tomography (PET) in head and neck squamous cell carcinoma (HNSCC) [11,12,13]. FLT is phosphorylated by TK1 and trapped within cells, allowing for noninvasive assessment of proliferative activity. Studies have demonstrated that FLT-PET can effectively monitor treatment response in HNSCC xenograft models and that early suppression of TK1 activity corresponds to decreased tracer uptake following radiation or EGFR-targeted therapy [13]. Although TK1 expression has been detected in OSCC tissue specimens, its functional role in regulating tumor invasion and metastasis has not been fully elucidated [11].

In this study, we performed an integrative analysis of multiple OSCC-related Gene Expression Omnibus (GEO) datasets to identify consistently dysregulated genes associated with metastasis. TK1 was identified as a key hub gene through GO and PPI analyses and was further validated using TCGA expression data and survival analyses. Functional assays demonstrated that TK1 knockdown reduced OSCC cell migration and cell movement capacity, supporting its role as a driver of OSCC progression and a potential therapeutic target.

## 2. Materials and Methods

### 2.1. Data Sources

Five microarray datasets (GSE9844, GSE23558, GSE30784, GSE37991, and GSE138206) were retrieved from the Gene Expression Omnibus (GEO; https://www.ncbi.nlm.nih.gov/geo/) (accessed on 12 May 2025) [14,15,16]. These datasets were selected based on sample size, data quality, and inclusion of both OSCC and adjacent normal oral tissues, allowing for robust identification of consistently differentially expressed genes (DEGs). The following five OSCC-related microarray datasets were obtained from the GEO database: GSE9844, GSE23558, GSE30784, GSE37991, and GSE138206. The platforms used for gene expression profiling were as follows: GSE9844, GSE30784, and GSE138206 were based on the Affymetrix Human Genome U133 Plus 2.0 Array (GPL570); GSE23558 utilized the Agilent-014850 Whole Human Genome Microarray platform (GPL6480); and GSE37991 was generated using the Illumina HumanRef-8 v3.0 expression BeadChip (GPL6883).

### 2.2. Identification of Differentially Expressed Genes

DEGs between tumor and normal samples were screened using GEO2R (https://www.ncbi.nlm.nih.gov/geo/geo2r/) (accessed on 12 May 2025), which implements the limma package in R. Genes with an adjusted *p*-value < 0.01 (Benjamini–Hochberg correction) and |log_2_ fold change| > 1 were considered significantly differentially expressed. Overlapping DEGs from at least three datasets were identified using InteractiVenn (https://www.interactivenn.net/) [17,18].

### 2.3. Gene Ontology Enrichment Analysis

GO enrichment analysis was conducted via ShinyGO (http://bioinformatics.sdstate.edu/go/) version 0.82. Enriched terms for biological process (BP), cellular component (CC), and molecular function (MF) categories were identified using thresholds of *p* < 0.01 and false discovery rate (FDR) < 0.05 [19,20,21].

### 2.4. Protein–Protein Interaction Network and Hub Gene Identification

The overlapping DEGs were submitted to the STRING database (https://string-db.org/) to generate a protein–protein interaction (PPI) network with a minimum interaction score of 0.4 [22,23]. The resulting network was imported into Cytoscape v3.10.2 for visualization and analysis. MCODE (v1.5.1) was used for module detection with default parameters (Degree Cutoff = 2, Node Score Cutoff = 0.2, K-Core = 2, Max Depth = 100). CytoHubba was employed to rank nodes based on five topological algorithms (Degree, EPC, MNC, DMNC, and MCC) [24,25]. A total of 15 top-ranked hub genes were identified and intersected across algorithms to obtain 13 consensus genes for further analysis.

### 2.5. Expression and Survival Analysis of Hub Genes

The Kaplan–Meier Plotter (https://kmplot.com/analysis/) (accessed on 16 May 2025) was used to evaluate the prognostic relevance of hub gene expression in OSCC [26,27]. Survival differences were analyzed using the auto-selected best cutoff with Cox regression modeling.

### 2.6. Analysis of Gene Expression Across Cancer Types and Disease Stages

To explore the expression patterns of candidate genes across diverse tumor types and stages, multiple publicly available platforms were utilized. The TIMER2.0 web server (http://timer.cistrome.org/) (accessed on 16 May 2025) was used to analyze gene expression differences between tumor and adjacent normal tissues across multiple cancer types within The Cancer Genome Atlas (TCGA), including head and neck squamous cell carcinoma (HNSC) [28,29]. For further confirmation at the protein level, immunohistochemistry (IHC) data from the Human Protein Atlas 24.0 (HPA 24.0) (https://www.proteinatlas.org/) (accessed on 16 May 2025) were examined to assess TK1 protein localization and expression intensity in both normal and HNSC tissues [30]. Additionally, UALCAN (https://ualcan.path.uab.edu/index.html) (accessed on 19 May 2025), an interactive portal for cancer transcriptome data analysis, was employed to investigate TK1 mRNA and protein expression levels across different pathological stages and lymph node metastasis statuses in HNSC patients [31,32].

### 2.7. Reagents and Antibodies

Primary antibodies against TK1 and GAPDH, along with HRP-conjugated secondary antibodies, were obtained from Genetex (Irvine, CA, USA). Ibidi culture inserts (Cat. No. 80209) were used for wound healing assays. All other reagents were purchased from Sigma-Aldrich (St. Louis, MO, USA).

### 2.8. Cell Culture and Migratory Subpopulation Enrichment

OSCC cell lines SCC4 (ATCC), and HSC3 (Sigma-Aldrich) were cultured in DMEM supplemented with 10% FBS, 100 U/mL penicillin, and 100 µg/mL streptomycin at 37 °C with 5% CO_2_. Migratory subpopulations were isolated by seeding SCC4 cells into Transwell inserts (8 µm pore size) and collecting cells that migrated to the lower chamber over repeated selection rounds to establish the M10 subline.

### 2.9. Transwell Migration Assay

Transwell inserts (8 µm pore) in 24-well plates were used to assess cell migration. Cells (3 × 10^4^) in serum-free medium were seeded in the upper chamber, and FBS-containing medium was added below. After 24 h, migrated cells were fixed, stained with 0.05% crystal violet, imaged, and quantified using ImageJ v1.52a.

### 2.10. Wound Healing Assay

For wound-healing assays, cells (3 × 10^4^) were seeded into two-chamber ibidi inserts. After 24 h, inserts were removed to generate a uniform gap. Cells were transfected with or without TK1 siRNA and allowed to migrate for 24 h. Wound closure was visualized at 0 and 24 h and quantified with ImageJ.

### 2.11. siRNA Transfection

TK1-targeting siRNAs and a non-targeting control were purchased from Sigma-Aldrich. Transfections were carried out using DharmaFECT 1 reagent (Horizon Discovery, Cambridge, UK) per manufacturer’s instructions. Cells were incubated for 24 h post-transfection prior to analysis. The siRNA sequence for TK1 was: 5′-GGUGAUCAAGUAUGCCAAA-3′.

### 2.12. Western Blotting

Total protein was extracted from cells and separated by SDS-PAGE, transferred onto PVDF membranes, and probed with primary antibodies (1:1000 dilution) overnight at 4 °C. HRP-conjugated secondary antibodies (1:10,000 dilution) were applied for 1 h at room temperature. Signals were detected using chemiluminescence (ECL) and imaged with a UVP system (Analytik Jena, Upland, CA, USA).

### 2.13. Statistical Analysis

Statistical comparisons of gene expression between OSCC and normal tissues (from TCGA) were performed using two-tailed *t*-tests. Multiple testing correction was applied using the Benjamini–Hochberg method. Kaplan–Meier survival analyses and univariate Cox regression were used to evaluate survival differences. For experimental data, statistical comparisons between two groups were performed using an unpaired two-tailed Student’s *t*-test. For comparisons involving more than two groups, two-way ANOVA followed by Sidak’s post hoc test was applied. Statistical significance was defined as *p* < 0.05.

## 3. Results

### 3.1. Identification of Consistently Dysregulated Genes in OSCC from Public GEO Datasets

To identify robust and clinically relevant differentially expressed genes (DEGs) associated with oral squamous cell carcinoma (OSCC), we selected five publicly available transcriptomic datasets from the GEO database: GSE9844, GSE23558, GSE30784, GSE37991, and GSE138206. Each dataset comprises gene expression profiles comparing OSCC tissues with adjacent or matched normal oral tissues. The sample distributions included 12 normal and 26 tumor samples in GSE9844, 5 normal and 27 tumor samples in GSE23558, 45 normal and 167 tumor samples in GSE30784, 40 paired normal and tumor samples in GSE37991, and 6 normal versus 5 tumor mucosa samples in GSE138206 (Figure 1F). We applied the GEO2R tool to each dataset to identify DEGs between tumor and normal samples. The DEGs from each dataset were visualized using volcano plots, highlighting significantly upregulated and downregulated genes in OSCC tissues (Figure 1A–E). The DEG counts varied across datasets: 615 in GSE9844; 2156 in GSE23558; 9966 in GSE30784; 7854 in GSE37991, and 847 in GSE138206. To identify consistently dysregulated genes across multiple datasets, we performed a Venn diagram analysis. This intersection revealed a total of 138 DEGs that were commonly altered across all five datasets (Figure 1G). These shared DEGs represent a core set of genes that may serve as reliable molecular indicators of OSCC pathogenesis and were thus prioritized for downstream functional and prognostic analysis.

### 3.2. Functional Enrichment and Identification of Hub Genes Among Consistently Dysregulated DEGs

To investigate the potential biological roles of the 138 DEGs commonly identified across the five GEO datasets, Gene Ontology (GO) enrichment analysis was performed using the ShinyGO v0.82 platform. In the biological process (BP) category, these genes were significantly enriched in pathways associated with cell proliferation, extracellular matrix organization, cytoskeletal structure, and signal transduction—processes frequently implicated in tumor progression and metastasis (Figure 2A). Analysis of the cellular component (CC) category further supported these findings, indicating that the 138 DEGs were predominantly involved in structural components of the cytoskeleton and extracellular matrix, including collagens and other matrix-associated proteins (Figure 2B). These results suggest that the shared DEGs may play important roles in regulating the physical and signaling architecture of OSCC cells. To explore molecular interactions and identify key regulatory genes, we constructed a protein–protein interaction (PPI) network using the STRING database. The network was imported into Cytoscape v3.10.2 for visualization and cluster analysis via the MCODE plugin (v1.5.1), revealing densely connected modules potentially representing functionally related gene clusters (Figure 2C). Subsequently, we applied the CytoHubba plugin to rank nodes within the PPI network using five topological algorithms: Degree, Edge Percolated Component (EPC), Maximum Neighborhood Component (MNC), Density of Maximum Neighborhood Component (DMNC), and Maximal Clique Centrality (MCC). The top 15 genes from each algorithm were cross-compared using a Venn diagram, which identified 13 hub genes consistently ranked across all five methods (Figure 2D). These 13 robustly identified hub genes included TRIP13, CHEK1, CDCA5, KIF4A, CENPE, AURKA, FOXM1, KIF14, ATAD2, TK1, CDK6, UHRF1, and KIF2C (Figure 2E). Several of these genes are known regulators of cell cycle progression, DNA replication, and chromosomal segregation, further supporting their potential relevance in OSCC pathogenesis and metastasis.

### 3.3. Expression and Prognostic Significance of Hub Genes in HNSC

To validate the clinical relevance of the identified hub genes, we examined their expression levels in head and neck squamous cell carcinoma (HNSC) using data from The Cancer Genome Atlas (TCGA) via the UALCAN portal. All 13 hub genes exhibited significantly elevated expression in tumor tissues compared to normal tissues (Figure 3), suggesting a potential role in tumor development and progression. To further assess their prognostic significance, Kaplan–Meier survival analyses were performed using the KM-Plotter tool, focusing on patients with stage II–III HNSC. Among the 13 genes, high expression levels of TK1 and CDK6 were significantly associated with poorer overall survival. Specifically, patients with high TK1 expression had a hazard ratio (HR) of 2.41 (log-rank *p* = 0.027), and those with high CDK6 expression had an HR of 1.95 (log-rank *p* = 0.014), indicating their strong association with adverse outcomes and more aggressive disease behavior (Figure 4). Based on these findings, TK1 emerged as a particularly compelling candidate for further investigation. We therefore selected TK1 as the primary focus for subsequent experimental validation to explore its role in OSCC progression and metastasis.

### 3.4. TK1 Is Overexpressed in Multiple Cancers and Correlates with OSCC Progression

To further validate the oncogenic relevance of thymidine kinase 1 (TK1), we examined its expression across different cancer types and clinical stages using multiple public databases. Analysis via the TIMER2.0 platform revealed that TK1 expression was markedly elevated in a wide range of cancer types compared to corresponding normal tissues, supporting its general association with tumorigenesis (Figure 5A). Consistent with the transcriptomic data, immunohistochemical (IHC) staining images from The Human Protein Atlas (HPA) demonstrated strong TK1 protein signals in human HNSC specimens, indicating elevated expression at the protein level (Figure 5B,C). Furthermore, expression data from the TCGA database showed that TK1 mRNA levels were significantly increased with advancing tumor stage and lymph node metastasis status in HNSC patients (Figure 5D,E). Protein expression data retrieved from the CPTAC database further confirmed this trend, demonstrating that TK1 protein levels were significantly higher in HNSC tissues compared to normal tissues, and progressively increased with tumor stage (Figure 5F,G). In parallel, expression profiling from the GEO datasets confirmed significantly elevated TK1 mRNA levels in OSCC samples relative to normal tissues (Figure 6), reinforcing the findings from TCGA and CPTAC. Collectively, these multi-platform analyses consistently indicate that TK1 is overexpressed in OSCC and correlates with tumor progression and lymph node metastasis. These results support the hypothesis that TK1 plays a pivotal role in OSCC malignancy and highlight the need for further functional validation to elucidate its role in metastasis.

### 3.5. TK1 Expression Promotes OSCC Cell Motility and Invasiveness

To investigate the potential role of TK1 in oral squamous cell carcinoma (OSCC cell motility, we further analyzed its expression in highly migratory OSCC subpopulations. The high-mobility SCC4 subline exhibited significantly elevated TK1 expression at the protein levels compared to the parental OSCC lines (Figure 7A), suggesting a strong association between TK1 expression and the migratory phenotype of OSCC cells. To functionally validate this association, we performed transient knockdown of TK1 using siRNA in OSCC cell lines and assessed cell motility through Transwell migration assays and wound-healing assays. Silencing TK1 expression significantly reduced the migratory capacity of OSCC cells in both experimental systems (Figure 7B,C), indicating that TK1 plays an active role in promoting OSCC cell movement. Furthermore, to assess the invasive behavior of OSCC cells, we conducted a Matrigel-coated Transwell invasion assay following TK1 knockdown. The results showed that TK1 silencing also significantly impaired the invasive capacity of OSCC cells (Figure 7D), supporting the hypothesis that TK1 contributes to both migratory and invasive properties associated with metastatic potential. Together, these in vitro findings demonstrate that TK1 is not only overexpressed in OSCC but also functionally contributes to the migratory behavior of cancer cells. These results support TK1 as a potential predictive biomarker for OSCC metastasis and a candidate therapeutic target for controlling tumor progression.

## 4. Discussion

Oral squamous cell carcinoma (OSCC) remains one of the most aggressive malignancies of the head and neck region, characterized by a high rate of local invasion, cervical lymph node metastasis, and frequent recurrence. Despite improvements in surgical techniques, radiotherapy, and systemic chemotherapy, the 5-year survival rate for OSCC patients remains stagnant at approximately 50%, largely due to late-stage diagnosis and the lack of effective molecular markers for early detection and therapeutic targeting [1,2,3,4]. Therefore, there is an urgent need to identify reliable biomarkers that not only predict disease progression but also offer potential for therapeutic intervention.

In this study, we adopted a systematic bioinformatics-guided approach to identify differentially expressed genes (DEGs) consistently dysregulated across five independent OSCC-related GEO datasets. Among the 138 overlapping DEGs, functional enrichment and protein–protein interaction analyses led us to a group of 13 hub genes highly associated with cell proliferation, extracellular matrix organization, and metastatic behavior. Among these, thymidine kinase 1 (TK1) emerged as a particularly promising candidate due to its consistent overexpression across datasets and significant correlation with poor survival in stage II–III head and neck squamous cell carcinoma (HNSC) patients. Although both TK1 and CDK6 were found to be significantly associated with worse survival in stage II–III HNSC patients, we selected TK1 as the primary focus for further investigation based on multiple factors. Firstly, in the TCGA pan-cancer analysis, CDK6 was relatively highly expressed in several normal tissues compared to tumor tissues in types such as BRCA, KICH, KIRC, LUAD, THCA, and UCEC, suggesting a potential limitation in its tumor specificity. In contrast, TK1 showed consistent and tumor-enriched expression across multiple cancer types and minimal expression in normal tissues—with the exception of KICH—supporting its relevance as a more cancer-selective biomarker. Secondly, although CDCA5 was also identified as one of the 13 hub genes and showed high expression in HNSC tissues, it was unexpectedly associated with better survival in patients with advanced-stage disease, which is contradictory to the oncogenic implication of its upregulation. This paradox led us to deprioritize CDCA5 as a potential driver of OSCC progression and metastasis. Based on this integrative assessment of expression specificity, survival relevance, and functional plausibility, we focused our downstream analyses on TK1 to evaluate its prognostic and biological role in OSCC malignancy. Subsequent experimental validation in OSCC cell lines confirmed that TK1 is highly expressed in highly migratory subpopulations and that its knockdown significantly impairs cell motility. These findings highlight TK1 as a potential predictive biomarker and therapeutic target in OSCC. In addition to its role in promoting migration, our Matrigel invasion assay further demonstrated that TK1 knockdown significantly impaired the invasive capacity of OSCC cells. These results, shown in Figure 7D, provide functional evidence that TK1 contributes to both cell motility and invasiveness, reinforcing its role in OSCC metastasis. This aligns with clinical observations from the TCGA and CPTAC datasets, where TK1 expression was positively correlated with lymph node involvement and advanced tumor stage. Together, these data strengthen the rationale for considering TK1 as a key driver of OSCC malignant progression.

TK1 is a cytosolic enzyme involved in the salvage pathway of DNA synthesis, catalyzing the phosphorylation of thymidine-to-thymidine monophosphate. As a marker of cell proliferation, TK1 has been extensively studied in various cancers, including lung, breast, colorectal, esophageal, and ovarian carcinomas. Elevated TK1 expression has been associated with poor prognosis, increased tumor burden, and higher metastatic potential [7,8,10,33]. For instance, in lung adenocarcinoma, TK1 overexpression has been shown to promote tumor growth and metastasis through Rho GTPase activation and downstream regulation of GDF15 [10]. Similarly, studies in breast and esophageal squamous cancer have shown that high serum or tissue levels of TK1 correlate with tumor progression and reduced survival [5,6].

While several of these findings are consistent with our observations, the present study provides novel insights by systematically integrating bioinformatics screening with experimental validation, specifically in the context of OSCC. Notably, few prior studies have investigated TK1 in OSCC. One autoantibody profiling study identified TK1 as an immunogenic protein in HNSC [1], and another imaging-based study linked TK1 expression to cellular proliferation using 18F-FLT PET imaging [11]. However, none have functionally demonstrated that TK1 contributes to OSCC metastasis through enhanced cell motility, as shown in our siRNA-based knockdown experiments. This represents a novel contribution to the field, providing the first direct functional evidence implicating TK1 in OSCC metastasis.

Although TK1 is broadly upregulated across multiple tumor types, its oncogenic role appears to be context dependent and varies by cancer type. In contrast to malignancies such as lung, breast, or colorectal cancer—where TK1-mediated pathways such as Rho GTPase or MAPK signaling have been implicated in tumor progression—relatively little is known about the mechanistic involvement of TK1 in OSCC. Given the limited understanding of molecular drivers of metastasis in OSCC and the lack of specific studies addressing TK1’s function in this cancer type, our study aimed to fill this gap. By integrating multi-dataset bioinformatic analyses with in vitro functional assays, we provide the first direct evidence that TK1 promotes cell motility in OSCC, suggesting its potential as a biomarker and therapeutic target uniquely relevant to this disease context. Although the TCGA-HNSC dataset includes multiple anatomical subsites, OSCC constitutes the vast majority (~90%) of HNSCC cases, and our GEO datasets and in vitro experiments were specifically derived from oral tumor tissues and OSCC cell lines, supporting the relevance of TK1 overexpression in OSCC [34,35].

The implications of our findings are significant. As a cell-cycle-regulated enzyme, TK1 could serve as both a prognostic biomarker and a therapeutic target, especially in high-risk OSCC cases where metastasis remains a clinical challenge [9,36,37,38]. Its overexpression in early to advanced stages of OSCC also raises the possibility of using TK1 as a diagnostic marker, either through tissue biopsies or serum assays [9,33]. In particular, the differential expression of TK1 in highly migratory OSCC cell subpopulations may reflect its role in epithelial–mesenchymal transition (EMT) or cytoskeletal remodeling, which warrants further investigation [39].

## 5. Conclusions

This study demonstrates that TK1 is consistently overexpressed in OSCC and plays a functional role in promoting tumor cell motility. Through integrative bioinformatics and in vitro validation, we highlight TK1 as a promising prognostic biomarker and potential therapeutic target for managing OSCC progression and metastasis.

## Figures and Tables

**Figure 1 diagnostics-15-01567-f001:**
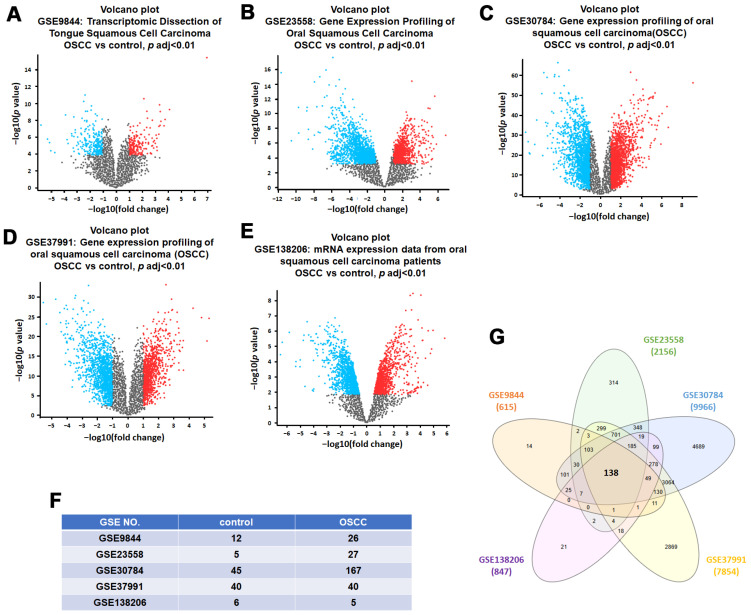
Identification of common differentially expressed genes (DEGs) in OSCC datasets. (**A**–**E**) Volcano plots showing upregulated (red), downregulated (blue), and non-significant (black) genes in five GEO datasets: GSE9844, GSE23558, GSE30784, GSE37991, and GSE138206. Adjusted *p*-value < 0.01 and |log2 fold change| > 1 were considered significantly differentially expressed. (**F**) Summary of sample size for each dataset. (**G**) Venn diagram illustrating common DEGs consistently altered across all five datasets.

**Figure 2 diagnostics-15-01567-f002:**
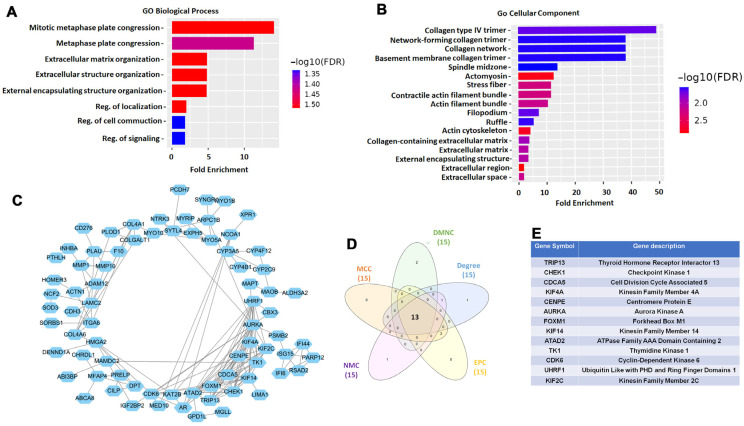
Functional enrichment and hub gene identification. (**A**) Gene Ontology (GO) analysis showing enrichment of biological processes among 138 DEGs. (**B**) Cellular component enrichment analysis among 138 DEGs. (**C**) Protein–protein interaction (PPI) network generated from STRING database and analyzed with MCODE in Cytoscape. (**D**) Venn diagram showing 13 common hub genes identified by five topological algorithms (Degree, EPC, MNC, DMNC, MCC) using CytoHubba. (**E**) List of hub genes.

**Figure 3 diagnostics-15-01567-f003:**
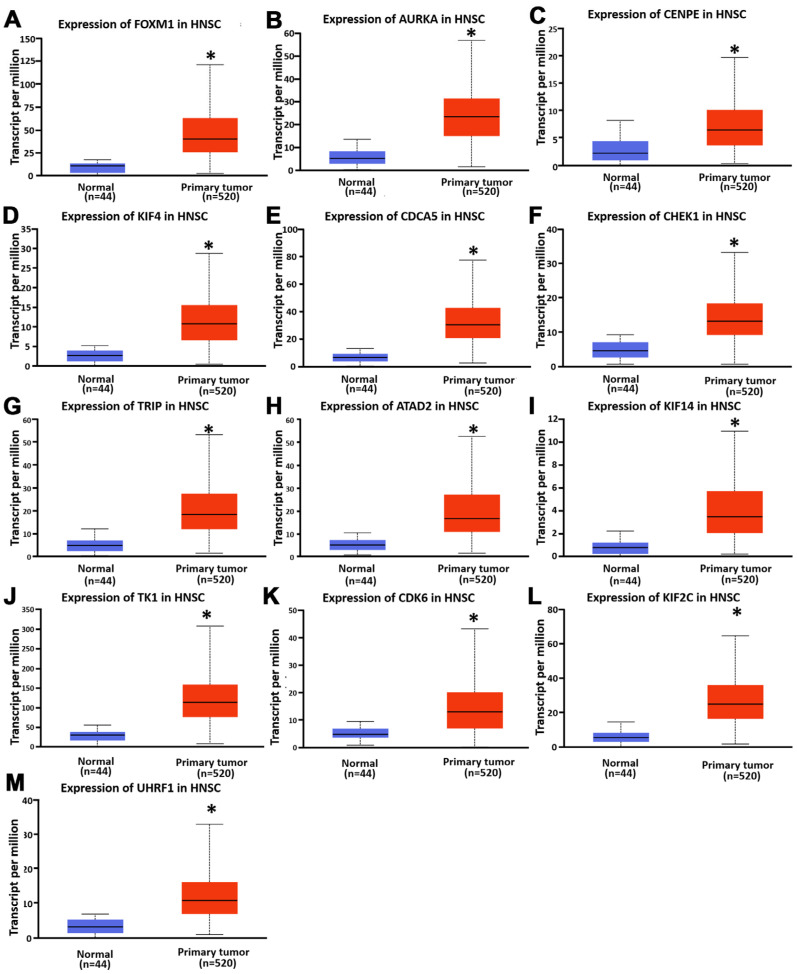
(**A**–**M**) Expression analysis of hub genes in HNSC. TCGA analysis showing mRNA expression of 13 hub genes in HNSC compared to normal tissues. * *p* < 0.05 compared to normal tissues.

**Figure 4 diagnostics-15-01567-f004:**
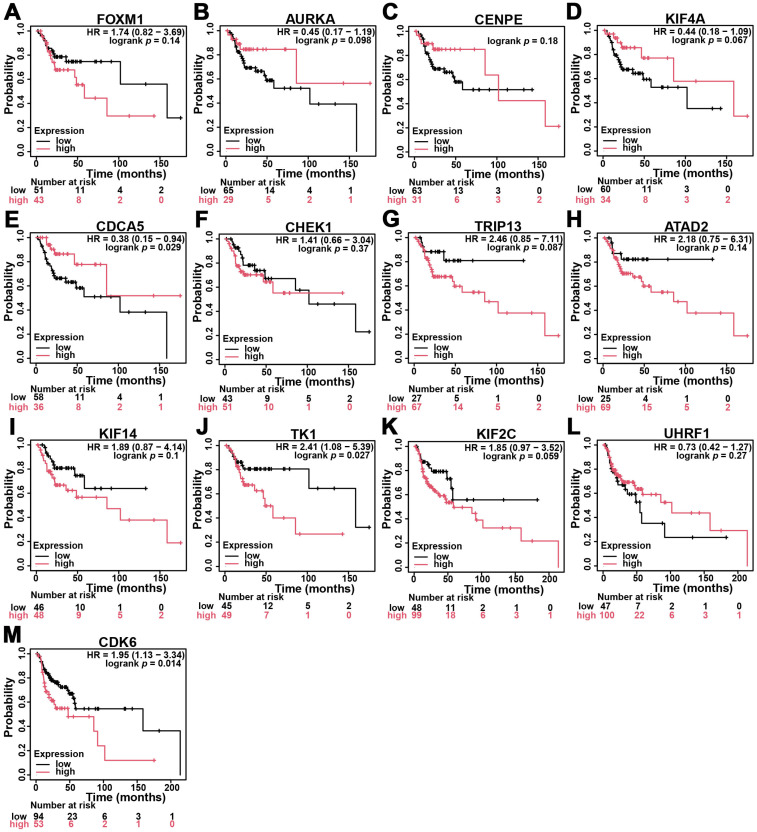
(**A**–**M**) Survival correlation of hub genes in HNSC. Kaplan–Meier plotter analysis of hub genes in stage II–III HNSC patients. Red represents higher expression, and black represents lower expression.

**Figure 5 diagnostics-15-01567-f005:**
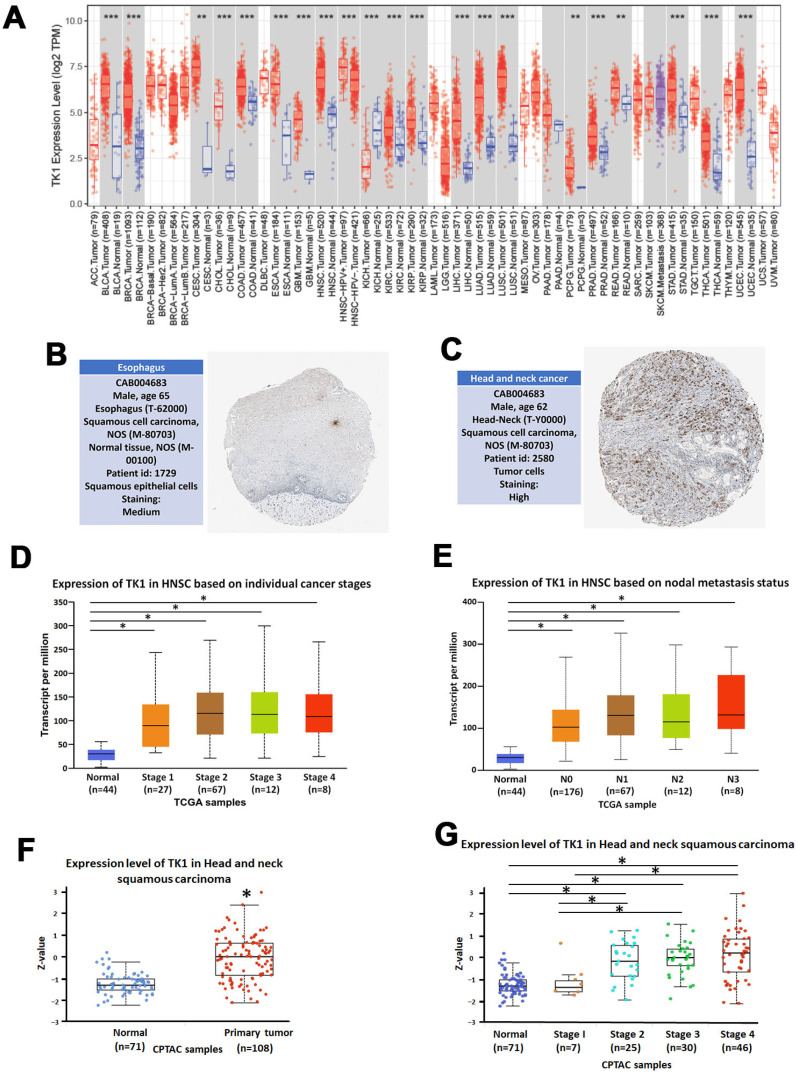
TK1 expression across cancers and stages. (**A**) TIMER2.0 analysis showing elevated TK1 expression across various cancer types. (**B**,**C**) IHC images from The Human Protein Atlas showing TK1 protein levels in normal and HNSC tissue. (**D**,**E**) TCGA data showing TK1 expression across clinical stages and lymph node grades. (**F**,**G**) CPTAC analysis reveals that TK1 protein levels in HNSC and cancer progression. * *p* < 0.05 compared to normal tissues, ** *p* < 0.01 compared to normal tissues, and *** *p* < 0.001 compared to normal tissues.

**Figure 6 diagnostics-15-01567-f006:**
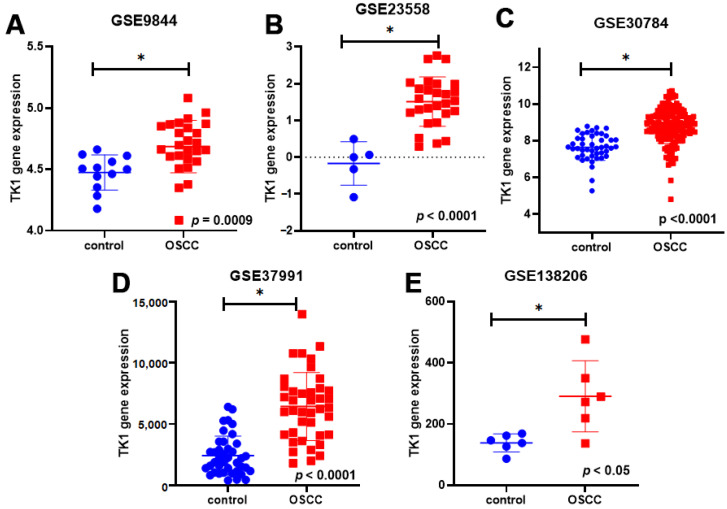
(**A**–**E**) Validation of TK1 expression in OSCC datasets. Dotplots of TK1 expression across OSCC samples and adjacent normal tissues from multiple GEO datasets, confirming consistent overexpression. * *p* < 0.05 compared to normal tissues.

**Figure 7 diagnostics-15-01567-f007:**
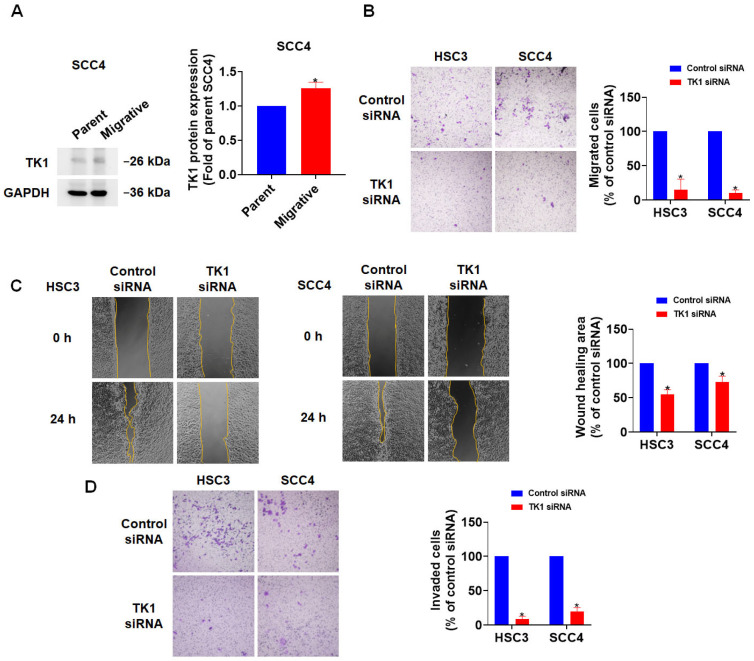
TK1 is associated with OSCC cell motility. (**A**) Western blot showing TK1 expression in parent SCC4 and high-mobility SCC4 sublines. (**B**) Transwell migration assay shows cell migration after siRNA-mediated TK1 silencing. (**C**) Wound healing assay shows the closure capacity in TK1-knockdown OSCC cells. (**D**) Transwell migration assay shows cell migration after siRNA-mediated TK1 silencing. * *p* < 0.05 compared to parent cell line and control siRNA.

## Data Availability

The data generated and analyzed will be made available from the corresponding author upon reasonable request due to the data used in this study are derived from publicly available datasets (e.g., GEO, TCGA), which are subject to platform-specific data use agreements. Any additional processed data are available upon request for academic purposes.

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
