# Peer review of "Thymidine Kinase 1 Expression Correlates with Tumor Aggressiveness and Metastatic Potential in OSCC"

_diagnostics, 2025, doi:10.3390/diagnostics15121567_

Round 1

Reviewer 1 Report

Comments and Suggestions for Authors

1. The information in lines 87-91 repeats the information in Figure 1F.
2. Figure 4E, J and M shows statistically significant differences in survival rates. However, high expression levels of TK1 and CDK6 were significantly associated with poorer overall survival, while for CDK5 it is the opposite. But then the authors select only TK1 from this list. Why? Explain this choice in the text.
3. Judging by Figure 5A, an increase in TK1 expression is common to almost all types of cancer, so why is it proposed to use it specifically for squamous cell carcinoma of the head?
4. If we compare Figure 5D and G, the data from different databases differ quite significantly, why?
5. We are talking about biopsy material in which TK1 expression was determined. Is there a prospect of using blood serum/plasma for these purposes? Will the trend of TK1 growth persist with progression, in your opinion?

Reviewer 2 Report

Comments and Suggestions for Authors

This is a well-structured and comprehensive study that investigates the role of thymidine kinase 1 (TK1) in oral squamous cell carcinoma (OSCC). By integrating data from multiple GEO datasets and validating findings with TCGA, CPTAC, and Human Protein Atlas resources, the authors convincingly demonstrate that TK1 is overexpressed in OSCC and correlates with tumor aggressiveness and metastasis. The in vitro functional assays further substantiate TK1’s role in promoting cell migration. The manuscript is scientifically sound, methodologically rigorous, and of potential clinical significance. Thus, I agree to accept this manuscript after several issues are addressed.

The authors validated the expression of TK1 in TCGA-HNSC dataset. As HNSC dataset include not only OSCC, but also tumors of other sites (e.g. larynx), the authors should clarify whether TK1 overexpression is specific to OSCC, or commonly observed among HNSCCs.

It is well known that a subset of HNSCCs are associated with human papilloma virus (HPV). The authors may clarify whether TK1 overexpression is associated with HPV infection status.

In Figure 5, the authors show increased levels of TK1 expression in HNSCC tumors with lymph node metastasis, suggesting that TK1 may promote metastasis. Additional experiments are needed to clarify this point, such as Matrigel invasion assay.

Round 2

Reviewer 1 Report

Comments and Suggestions for Authors

I have no more comments on the manuscript.